



# Process-based microphysical characterization of a strong mid-latitude convective system using aircraft in situ cloud measurements

Mireia Papke Chica[1,2,7], Valerian Hahn[1,2], Tiziana Braeuer[1], Elena de la Torre Castro[1,2], Florian Ewald[1], Mathias Gergely[3], Simon Kirschler[1,2], Luca Bugliaro Goggia[1], Stefanie Knobloch[1], Martina Kraemer[2,4], Johannes Lucke[1,5], Johanna Mayer[1], Raphael Maerkl[1,2], Manuel Moser[1,2], Laura Tomsche[1,2], Tina Jurkat-Witschas[1], Martin Zoeger[6], Christian von Savigny[7], Christiane Voigt[1,2]

[1] Institute of Atmospheric Physics, German Aerospace Center, DLR, Oberpfaffenhofen, Germany

[2] Institute of Atmospheric Physics, Johannes Gutenberg University Mainz, Mainz, Germany

[3] Deutscher Wetterdienst, Hohenpeissenberg Meteorological Observatory, Hohenpeissenberg, Germany

[4] Forschungszentrum Jülich GmbH, Jülich, Germany

[5] Faculty of Aerospace Engineering, Delft University of Technology, Delft, Netherlands

[6] Institute for Flight Experiments, German Aerospace Center, DLR, Oberpfaffenhofen, Germany

[7] Institute of Physics, University of Greifswald, Greifswald, Germany

*Correspondence to*: Christiane Voigt (Christiane.Voigt@dlr.de)

**Abstract.**

Clouds in the mixed-phase temperature regime impose a large uncertainty onto climate prediction models, in part due to incomplete knowledge of the degree of glaciation affecting cloud radiative properties. To achieve a better representation of these clouds, it is crucial to improve the understanding of ice nucleation and growth as well as microphysical properties determining the cloud phase. In this case study, we provide a rare data set of aircraft in situ measurements in a strong mid-latitude convective system extending from the boundary layer to the tropopause and aim to extend the sparse database of such measurements. Data were obtained with the research aircraft HALO and cloud properties were probed with the Cloud and Aerosol Spectrometer (CAS-DPOL) and the Cloud Imaging Probe grayscale (CIPg) during the CIRRUS-HL mission above Southern Germany in July 2021. Microphysical properties of the convective cloud system were measured along a 58-minute stepwise descent between the ground weather stations of Hohenpeissenberg and Munich at temperatures of -35 °C, -23 °C, -13 °C, -7 °C, and -1 °C. A phase identification (liquid/ice) of particles with diameters > 50 μm was achieved using the particle images of the CIPg. Based on recent work, clouds were categorized into four groups with different microphysical properties: *Mostly Liquid*, *Coexistence*, *Secondary Ice*, and *Large Ice*. High concentrations of large ice crystals were observed in upper layers at temperatures between -35 °C and -13 °C, confirming the importance of the Wegener-





Bergeron-Findeisen process for mid-latitude convection. Exceptionally high vertical motions for mid-latitudes of up to +/- 4 ms$^{-1}$ encountered in the convection promote various freezing and ice growth processes, which in this system led to high ice water contents of up to ~ 1.2 gm$^{-3}$ and to instrument icing. In contrast, low-level clouds near -1 °C encountered at lower vertical velocities were predominantly composed of liquid droplets and contained precipitated large ice in low concentrations.

We find that mechanisms initiating ice nucleation and growth strongly depend on temperature, relative humidity, and vertical velocity and variate within the cloud system. Our measurements represent a unique in-flight data set on microphysical cloud properties of a strong midlatitude convective event and invite for detailed cloud model evaluations and radar intercomparisons with focus on the mixed-phase temperature regime.

## 1    Introduction

Understanding the behavior of clouds requires insight into processes from the large-scale meteorological environment down to small-scale microphysics. These microphysical processes determine a cloud's microstructure and particle microphysics which in return affect the cloud climate impact. In terms of radiative properties, the degree of glaciation plays an important role (Noh et al., 2013). This is especially of interest for clouds in the mixed-phase temperature regime between -38 °C and 0 °C, where liquid and ice particles can coexist. Cloud liquid of mixed-phase clouds (MPC) in this temperature regime tends

to be underestimated in climate models (Nam et al., 2014; Zhang et al., 2018), which is mainly due to often misrepresented microphysical processes taking place and determining the cloud's phase composition. A false representation of the phase composition alone can make a difference of 17 Wm$^{-2}$ in the radiative effect when it is assumed to be completely liquid versus fully composed of ice crystals (Lohmann et al., 2016a).

Observations of MPCs are difficult to obtain due to sparse spacial and temporal coverage (Tan et al., 2016; Jensen et al., 2016),

which is especially the case for convective clouds (Mülmenstädt et al., 2015). As of now, most airborne measurements of convective MPCs are conducted in tropical and subtropical latitudes. For instance, Jäkel et al. (2017) conducted measurements in deep convective clouds above the Brazilian rainforest, determining the extent and phase composition of the mixed-phase layer in clean and polluted conditions. Heymsfield et al. (2005) analyzed homogeneous ice nucleation in deep tropical convection and how large particles influence its efficacy. An analyis of mycrophysical processes in Amazonian deep

convective clouds in clean and polluted environments in the gamma phase space was conducted by Cecchini et al. (2017). Fewer studies evaluate MPCs in mid-latitude convective systems, e.g., Carey et al. (2008) in altocumulus clouds. Jensen et al. (2016) sampled a variety of deep convective events in South-central Oklahoma and compared them to radar reflectivities. Noh et al. (2013) evaluated the vertical distribution of ice and liquid water content (*LWC*) in mid-latitude MPCs and distinguished between three different cloud types and Huang et al. (2021) conducted measurements of wintertime shallow convective MPCs

over the Southern ocean. Generally, evaluations of MPCs are more common for stratiform clouds (Korolev et al., 2007b; Costa et al., 2017). As an example, Hogan et al. (2002) probed stratiform clouds embedded in convective systems from aircraft and polarimetric radar in temperatures between -5 °C and -11 °C, also encountering the Hallett-Mossop process (Sect. 2.5.3).



Further, Boudala et al. (2004) parameterized the liquid fraction of stratiform clouds in high and mid-latitudes over 10-second intervals, detecting an increase of the liquid fraction with rising temperatures and a liquid fraction minimum between -10 °C

and -20 °C. A recently published study invesitages aersosol-cloud interactions and optical properties of shallow supercooled stratiform clouds via ground based remote sensing techniques (Radenz et al., 2021). In Lee et al. (2021) microphysical interactions of cloud particles and aerosols, as well as microphysical processes leading to the development of mixed-phase stratocummulus clouds above the Seoul area were evaluated using a large-eddy simulation framework to gain deeper understanding stratiform MPCs.

While the majority of data represented in climate models is collected using remote sensing, e.g., satellite, lidar, or radar retrievals (Heymsfield et al., 2018; Hogan et al., 2003; Hu et al., 2010), for this work, data are collected in situ. Airborne measurements as presented here support the further development of radar retrievals as they supply detailed information about the clouds' microphysical properties and processes in less well-understood mixed-phase layers of convective systems (Trömel et al., 2021). Furthermore, bulk measurements, i.e., the synergy of in situ and remote sensing techniques, will lead to an

advanced understanding of globally distributed MPCs (Korolev et al., 2017). The exceptional insight into the clouds' compositions and particle forms, as well as direct conclusions about real-time microphysical processes provided by in situ measurements, will further help to verify and substantiate climate models. It is therefore crucial to widen the database for in situ measurements in mixed-phase convective clouds.

The aim of this study is to provide a comprehensive overview of the influence microphysical processes exert on the phase

composition of mid-latitude convective clouds for the full temperature range at which MPCs can be encountered. For this purpose, deep convective clouds are sampled in summertime over land between -36 °C and -1 °C, featuring unusually high vertical velocities ($W$) for mid-latitudes ($\pm$ 4 ms$^{-1}$) that are comparable to convective conditions in the tropics (Costa et al, 2017). It should be noted that in the core of the convective system vertical velocities might have been much higher, but could not be probed without compromising flight safety. In addition to vertical velocities, similar values for *LWC, RH, T* and *N* are

also found in tropical MPC measurements conducted by Cecchini et al. (2017).

While there are many approaches for identifying and characterizing MPCs, summarized in Korolev et al. (2017), in this work, clouds are defined and characterized based on a process-based approach introduced by Costa et al. (2017). While Costa's classification of MPCs in mid-latitudes mainly focuses on stratocumulus and stratus clouds (low-level stratiform clouds) in this work, we measure and classify deep convective cumulus congestus clouds. Costa (2018) serves as a good reference, since

data were collected with the NIXE-CAPS instrument (Krämer et al., 2016) containing a CAS-DPOL (Baumgardner et al., 2001) and a CIPg, both of which are used in this work as well.

The cloud classification adapted from Costa et al. (2017) is based on the degree of glaciation, which is determined by the prevailing microphysical processes (Fig. 1). The first distinction (Type 1 or 2) is based on the mass modes of a particle size distribution (PSD). Here, the presence of a second mode in the PSD indicates high concentrations of ice particles with diameters

($D_\mathrm{p}$) larger than 50 µm and thus a *Large Ice* cloud. While in Costa et al. (2017) Type 2 clouds are also referred to as *Wegener-Bergeron-Findeisen/Large Ice* named after the Wegener-Bergeron-Findeisen process (WBF) (Wegener, 1912; Bergeron,


1928; Findeisen W., 1938), in this study, we base the name on the distinctive cloud particle composition, hence *Large Ice*. Next, Type 1 is subdivided into three groups: *Mostly Liquid*, *Coexistence,* and *Secondary Ice*. The second distinction is more elaborate and is based on further microphysical properties derived by images of the CIPg and scattering signals of the CAS-

DPOL. Furthermore, ambient conditions, above all temperature, $W$ and relative humidity with respect to water $RH_w$ and ice $RH_i$ are a crucial part of the differentiation, as they strongly influence cloud microphysical processes (Korolev, 2007b).

In more detail, *Mostly Liquid* clouds are characterized by mainly containing small liquid droplets ($D_p < 50$ μm) and low concentrations of large aspherical particles ($D_p > 50$ μm) (Table 1). Therefore, the PSD solely displays one mass mode, as can be seen in Fig. 12. If large particles are present, they most likely sedimented from upper cloud layers. As expected, in this

study a *Mostly Liquid* cloud was probed at higher temperatures just below freezing and with $RH_w$ near 100 % (Sect. 2.5.2).

With decreasing temperatures, it is more likely for *Coexistence* clouds to occur. Here, liquid droplets coexist with a larger fraction of ice particles, which mostly formed in situ versus through sedimentation as in *Mostly Liquid* clouds. Number concentrations of small droplets (liquid or frozen) still dominate the PSDs, but large ice particles appear more often. Although PSDs typically display one mass mode for $D_p < 50$ μm deviations are found in this work. In this cloud type the freezing process

already started, but $RH_w$ and $RH_i$ remain > 100% due to strong vertical updrafts (Costa et al., 2017).

*Secondary Ice* clouds emerge as a result of secondary ice production at lower temperatures. They contain high concentrations of small hydrometeors, which most likely are frozen droplets and irregular shaped ice particles. Similar to *Coexistence* clouds,

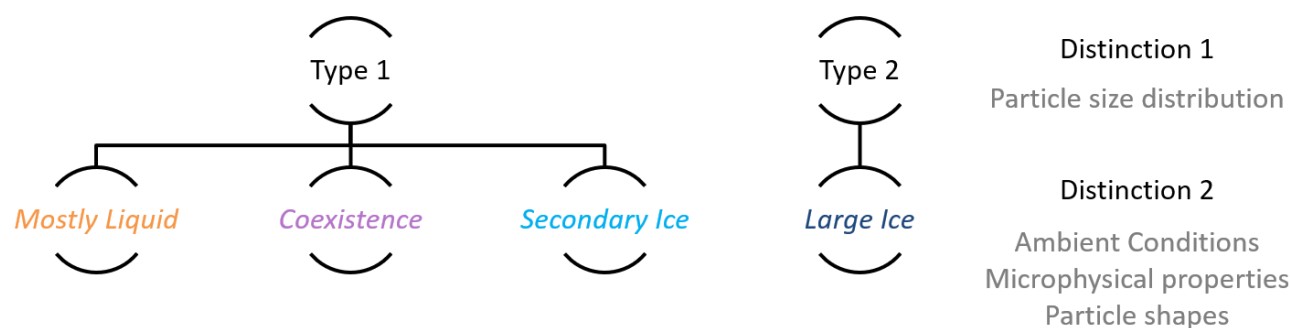

**Fig. 1: Schematic of cloud classification in the mixed-phase temperature regime based on the degree of glaciation. This approach was first introduced in Costa (2018) and Costa et al. (2017) for low-level clouds above the UK. Here, the classification is applied for deep convective clouds above Southern Germany.**





PSDs are expected to display a clear mass mode for particles with $D_p < 50\,\mu m$. In this work however, deviations are found as well. At higher $W$ and consequently a more turbulent environment, processes such as riming or splintering also referred to as

the Hallett-Mossop process (Hallett and Mossop, 1974), as well as drop freezing (Lawson et al., 2015) or ice-ice collisions (Yano and Phillips, 2011) take place producing secondary ice particles. Furthermore, contact freezing, i.e., ice crystals colliding with supercooled water droplets causing them to freeze and contributing to an advanced glaciation. Contact freezing of supercooled droplets can also lead to icing on the instrument's housing, which occurred during the evaluated flight (Sect. 2.4).

**Table 1: Conditions for differentiating between the cloud types in the mixed-phase temperature regime. While the first distinction (Fig. 1) is based on the mass modes of the particle size distribution (PSD), the second distinction is based on ambient conditions and microphysical properties.**

| | | Type 1 | | | Type 2 |
|---|---|---|---|---|---|
| | | Mostly Liquid | Coexistence | Secondary Ice | Large Ice |
| Distinction 1 | Number of PSD mass modes | 1: $D_p < 50\,\mu m$ | 1: $D_p < 50\,\mu m$ | 1: $D_p < 50\,\mu m$ | 2: $D_p < 50\,\mu m$ and $D_p > 50\,\mu m$ |
| Part of Distinction 2 | $N_{small}$ ($D_p < 50\,\mu m$) | > 1 cm⁻³ | > 1 cm⁻³ | > 1 cm⁻³ | < 1 cm⁻³ |
| | $N_{large}$ ($D_p > 50\,\mu m$) | Few, but present. (Drizzle drops or ice crystals) | Present (Ice crystals) | Present (Ice crystals) | Present in high concentrations (Ice crystals, graupel) |

Finally, at lower $W$ the $RH_w$ might drop below 100% and $RH_i$ remain above saturation, initiating the WBF process in Type 2

clouds. Here, ice crystals grow at the expense of liquid particles finally leading to fully glaciated clouds, which feature large ice crystals and relatively small concentrations of small particles. In these clouds concentrations of small particles are expected to lie below 1 cm⁻³ (Table 1) (Costa et al., 2017) and while high concentrations of large ice particles are present, hydrometeors with $D_p < 50\,\mu m$ still dominate the PSD.

In the scope of this work, measurements are obtained during the course of the CIRRUS-HL campaign (Sect. 2.1) with the High

Altitude and Long-Range Research Aircraft (HALO) (Krautstrunk and Giez, 2012). Data of the CAS-DPOL and the CIPg (Sect. 2.2) mounted on the wings of HALO are evaluated. Convective clouds probed above Southern Germany are analyzed regarding their microphysical properties, ice crystal shapes and processes that might have taken place within the cloud leading to the probed particle composition (Sect. 2.5). In Sect. 2.6 the results are compared to previous classifications of clouds in the mixed-phase temperature regime in mid- and tropical latitudes by Costa et al. (2017) and implications for mid-latitude

convection are discussed. Finally, Sect. 3 summarizes the study and provides an outlook.





## 2 Campaign and instrumentation

### 2.1 CIRRUS-HL

The CIRRUS-HL campaign on cirrus in high latitudes, led by the German Aerospace Center (*Deutsches Zentrum für Luft-und Raumfahrt*) (DLR) and the Johannes Gutenberg University Mainz, is a joint atmospheric research project by German research centers and universities within the Deutsche Forschungsgemeinschaft (DFG) HALO SPP 1294 framework (https://cirrus-hl.de/; http://www.halo-spp.de/). The campaign is a successor of the ML-CIRRUS campaign (Voigt et al., 2017) and the ACCRIDICON campaign (Wendisch et al., 2016). During this HALO mission, in situ and remote sensing cloud instruments

were combined, providing insights into nucleation, properties, and climate impact of ice clouds in high and mid-latitudes in June and July 2021. While the project focused on probing cirrus clouds induced by different meteorological regimes, convective and mixed-phase clouds were probed as well. On 8 July 2021, the flight plan (Fig. 2) included flight legs over

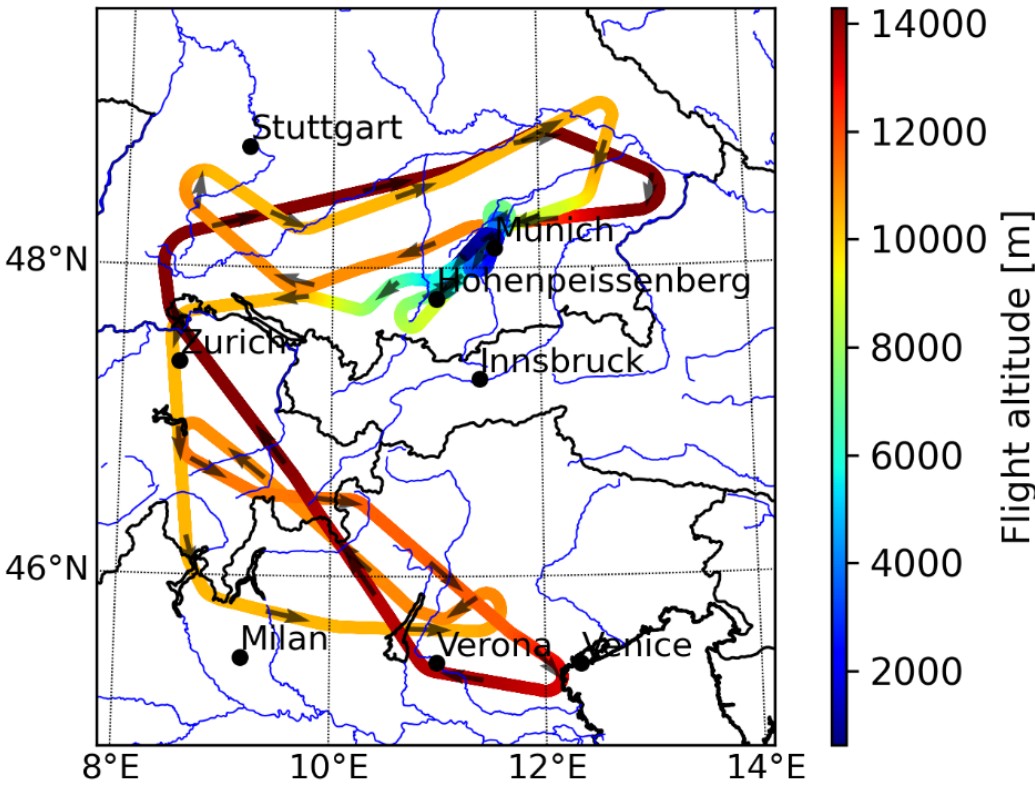

**Fig. 2: Flight track of HALO on 8 July, 2021, between 13:12 UTC and 18:04 UTC. The evaluated flight section is limited to a shuttle over the Meteorological Institute Munich and the climate reference station Hohenpeissenberg Meteorological Observatory between 17:00 UTC and 17:55 UTC. The colorbar displays the flight altitude. Values for the altitude were obtained from the Basic HALO Measurement And Sensor System (BAHAMAS).**



Southern Germany, Switzerland and Northern Italy. Figure 2 showcases the flight track color coded with the flight altitude.
Measurements evaluated for this work are limited to a 58-minute flight section, as opposed to the whole flight, shortly before
the final descent back to Oberpfaffenhofen. The reason for doing so is threefold. First, in this selected section, two radar
stations were passed providing additional and valuable information about the meteorological situation, second, according to
reports by the flight crew, the convective system was particularly strong and therefore of interest in terms of MPCs, and finally,
decreasing the data load for evaluation allowed for a comprehensive analysis of the clouds. These MPCs displayed in Fig. 3
were probed during a shuttle over the Meteorological Institute Munich (MIM) of the Ludwig-Maximilian university (LMU) at
flight levels (FL) 280, 230, 180 and 150 corresponding to 8873 m, 7303 m, 5707 m and 4783 m altitudes, as well as during a
profile over the climate reference station Hohenpeissenberg Meteorological Observatory (MOHp) at FL 150 and 120 (3840 m).
Both, MOHp and LMU operate radars.

**2.2    The CAS-DPOL and the CIPg**

For this campaign, HALO was equipped with eight wing probes (Voigt et al., 2017). For evaluation, two of the eight wing
probes were combined to gain a wider particle size range. The Droplet Measurement Technologies (DMT) light scattering
spectrometer, Cloud Aerosol Spectrometer with Depolarization Unit (CAS-DPOL) (Baumgardner et al., 2001; Voigt et al.,

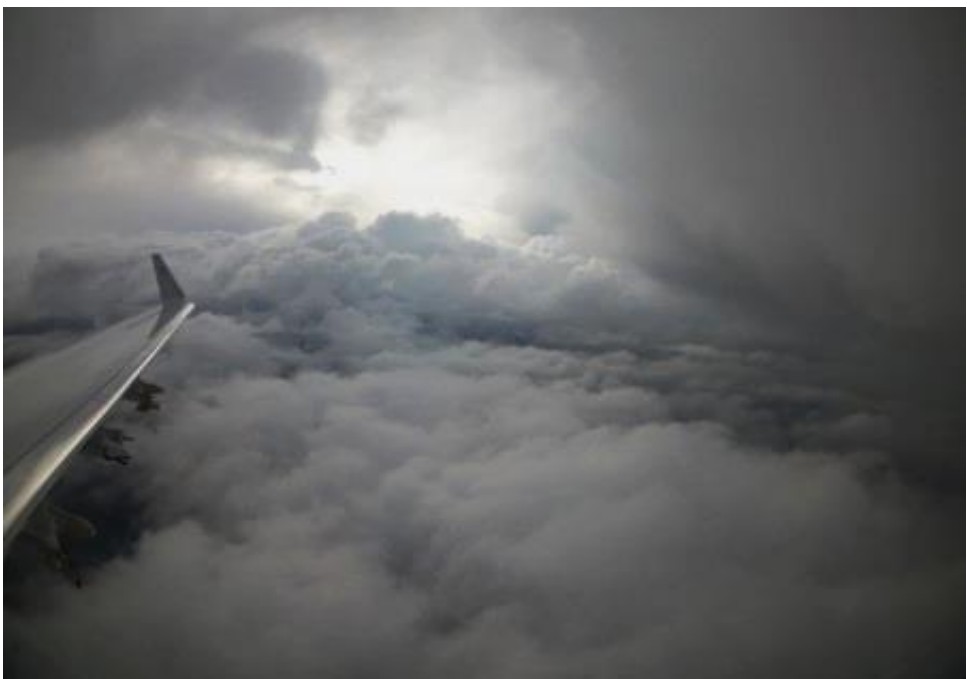

**Fig. 3: Image displaying convective clouds as well as the left wing of the HALO
aircraft with attached cloud probes. Photo was taken by a GoPro camera during the
CIRRUS-HL flight with HALO above Southern Germany on 8 July 2021 at 17:44:30
UTC [Image provided by the University of Leipzig].**



2021) was positioned at the outer left wing and the Grayscale Cloud Imaging Probe (CIPg), as part of the Cloud Combination Probe (CCP) (Weigel et al., 2016) was mounted onto the outer right wing of the aircraft. Even though these instruments were located on opposite sides, similar measurement conditions can be assumed.

In CAS-DPOL, light from a laser beam is scattered by individual particles (0.5 µm to 50 µm) passing through the beam. The forward-scattered light is registered by a photo detector after passing through collecting optics and a beam splitter (Kleine, 2019). Particle sizes are determined by assuming Mie Scattering theory with additional assumptions of spherical water droplets, a known particle refractive index and wavelength of the incident light (0.658 µm), as well as a particle size range complying with the Mie-regime. In addition to the forward scattered light detector, CAS-DPOL also contains a backward scattering light

detector, which allows for a differentiation between spherical and aspherical particles based on the assumption that liquid droplets are spherical and ice crystals have aspherical shapes (Järvinen et al., 2016). In this work, only the forward detection of particles is analyzed. All particles in the CAS-DPOL size regime are therefore assumed to be spherical. This is justified, due to the fact that in clouds, where CAS-DPOL number concentrations are particularly high, e.g., *Mostly Liquid* and *Coexistence*, particles tend to be liquid and therefore spherical. In *Large Ice* clouds, most particles with $D_\mathrm{p} < 50$ µm are frozen,

nonetheless number concentrations in this size range are much lower. Finally, in *Secondary Ice* clouds a distinction proves to be more difficult for $D_\mathrm{p} < 50$ µm. However, it was succeeded by evaluating the influence of acting microphysical processes via $W$, $RH_\mathrm{i}$, $RH_\mathrm{w}$ as well as $T$ and additionally taking into account an aspherical fraction analysis by Costa et al., (2017). Furthermore, only particles $> 3$ µm were evaluated, since particles with smaller diameters are referred to as aerosols (Costa et al., 2017; Righi et al., 2020).

The CIPg detects particles in a size range between 7.5 µm and 960 µm by capturing the shadow image of particles passing through a collimated laser beam from a 45 mW, 0.658 µm wavelength diode laser on a linear array of 64 diodes. A particle is detected when the light level of a diode is decreased by at least 25 % (Baumgardner et al., 2001). In order to minimize uncertainties in the detection of a particle, a depth of field correction (Korolev, 2007a) was applied with the OASIS software licensed by DMT (Droplet Measurement Technologies, 2013). The registered changes in the photo detectors are stored at a

rate consistent to the probe airspeed and size resolution of 15 µm. Furthermore, the CIPg images particles with three gray-scale levels allowing a higher sensitivity to light and therefore an easier detection of out of focus particles (Luebke et al., 2016; O'Shea et al., 2021; O'Shea et al., 2019). The CIPg raw data files are then processed further with the OASIS.

   While the size ranges of the two instruments complement each other, they also overlap, requiring a transition threshold between the instruments. Even though the sizing resolution in the upper CAS-sizing bins is higher than in the lower CIPg-sizing bins,

it is known that better particle sizing statistics in this range are provided by the CIPg, since its sample volume is larger (Luebke et al., 2016). Therefore, a transition threshold was chosen at 29.5 µm.

   The evaluation of MPCs crucially depends on a method to discriminate ice particles from liquid droplets, as this coexistence not only distinguishes MPCs from other cloud types, but also indicates in what stage the cloud was sampled. A particle phase identification (liquid/ice) was conducted by analyzing the images taken by the CIPg based on two assumptions: 1) ice particles

have irregular and non-circular shapes, therefore images differing from circles indicate ice; 2) liquid particles provide circular



images due to their spherical shapes (Korolev et al., 2017). For this work, it is assumed that the CIPg is able to distinguish spherical and non-spherical particles with $D_p > 50$ µm. This threshold is applied during processing with the OASIS software. The exact discrimination of frozen and liquid droplets, especially for particles with $D_p < 5$ µm, however, remains an unresolved and highly debated challenge in the scientific community (Korolev et al., 2017; Järvinen et al., 2016).

## 2.3     Methods and uncertainties

Various studies have been performed, during this and previous campaigns, to validate the measurement accuracy of the deployed instrumentation. For instance, CAS-DPOL data have been compared to observations by the CCP and the Cloud Droplet Probe (CDP) in tropical convection (Andreae et al., 2018; Braga et al., 2017a; Braga et al., 2017b) and an agreement better than 20% in $N$ and particle sizes has been found (Braga et al., 2017a). Furthermore, a good agreement was discovered between CAS-DPOL and CDP data in low level clouds in Africa (Taylor et al., 2019). To improve the quality of the results of this study instrument-related measurement uncertainties, i.e., Mie ambiguities (Borrmann et al., 2000; Kleine, 2019), shattering (Korolev and Field, 2015), coincidence (Lance et al., 2010), and the Ram Rise Effect (Weigel et al., 2016) were analyzed and eliminated.

Due to the ambiguous resonance structure of the water Mie curve for small particle sizes, values of a detected scattering cross section might be assigned to multiple particle diameters. To eliminate this bias, predefined bins with non-differentiable bin edges were combined into single Mie size bins.

Another possible uncertainty is induced by the bursting of ice particles once they hit the probe's surface which is referred to as shattering (Korolev and Field, 2015). Once shattered particles enter the instrument's sample volume small particle concentrations may be overestimated. Whether shattering is present or not, can be determined utilizing an "Inter Particle Arrival Time" (IPT) analysis (Field et al., 2003; Kleine, 2019). For the optical array probe CIPg this analysis was automatically applied during processing and shattered particles, if present, were sorted out based on a correction by Korolev and Field (2015). In case of CAS-DPOL, the IPT analysis was carried out manually using the particle by particle file and resulted in no apparent and distinguishable shattered fragments in most of the flight layers. This might be due to the fact that shattered particles either lie in the same size range as intact particles or shattering was not high enough to cause a significant measurement uncertainty. In accordance with Korolev and Field (2015), images by the CIPg were used as reference to determine, in which section of the flight shattering might have occurred the most. Since it is expected that due to differing inlet types CDP counts less shattered particles than CAS (Braga et al., 2017a), the instruments' number concentrations between 0.001 cm$^{-3}$ and 10 cm$^{-3}$ for particle sizes between 3 µm and 30 µm were compared. Deviations were found in the first two layers, indicating that shattering might have occurred at the coldest temperatures. As a result, all particles with IPT < 0.002 s were eliminated from the evaluation of the affected layers. A correction for shattered particles inevitably implies sorting out particles based on the assumption of them not being intact. Since filtering out particles can lead to an elimination of qualified particles and subsequently to an



underestimation of particle concentrations, and due to unclear results of the IPT analysis at the remaining layers, no further measures were taken for the CAS-DPOL size range (3 µm to 29.5 µm).

High values of $N$ provoke small spatial distances between particles traversing the laser beam simultaneously, which
complicates the differentiation of the scattered light of each particle individually and leads to a systematic bias referred to as coincidence (Lance et al., 2010). The influence of coincidence was estimated based on the works of Lance (2012) and Kleine (2019). According to Lance (2012) systematic errors due to concidence, e.g., an undercounting bias of 27 % and an oversizing bias of 20 % to 30 %, can be reduced by modifiyng the probes' optics. Kleine (2019) applies the same method and compares the $N$s of a modified and unmodified CAS-probe in cross calibration flights. His results, too, demonstrate that a smaller sizer
pinhole, as part of the optical set up, reduces coincidence biases in the extended sample area. Therefore, CAS-DPOL was modified with a 600 µm pinhole to reduce the influence of erroneously qualified particles within the extended sample area (Lance et al., 2010). Lance (2012) demonstrates that the effect of coincidence is negligible at $N < 1100$ cm$^{-3}$ for a modified CDP with a pinhole of 800 µm. In Molleker (2014) a different method was used, but the results illustrate that a CAS with a 500 µm  pinhole provides no indication of coincidence at N $< 1500$ cm$^{-3}$. Both of these instruments have very similar optical
setups and probe housings compared to the CAS-DPOL. Additionally, these two analyses serve as a good reference, since CAS-DPOL was modified with a pinhole that falls between that of the CDP and CAS. Since maximum $N$s of only 200 cm$^{-3}$ were reached and based on the conclusions of the aforementioned two studies, no correction of coincidence was necessary in this work. While Mie ambiguities mainly affect the correct sizing of particles, shattering and coincidence can affect both, the sizing and  counting of particles. With no corrections applied, an uncertainty related to Mie ambiguities is propagated $< 10$ %
for particle sizes between 10 µm and 30 µm in McFarquhar et al. (2017) and for counting biases an uncertainty between 10 % and 30 % is estimated. Since corrections were applied or artefacts could not be detected, an uncertainty of $< 10$ % is estimated for the CAS-DPOL size range.

Finally, a closer look was taken at the Ram Rise Effect, which in this case refers to an increase in pressure and temperature compared to ambient conditions, due to an air compression in the probe's detection region. According to Weigel et al. (2016)
associated thermodynamic effects are common in open path forward front instruments such as the CIPg and result in an overestimation of particle concentrations. In order to receive a corrected $N$, the measured concentration is multiplied with a correction factor $\xi$, for aircraft velocities between 60 ms$^{-1}$ and 250 ms$^{-1}$. This factor lies between 0.90 and 0.99 for the evaluated flight section and strongly depends on the aircraft's true air speed, i.e., the accuracy of uncorrected measured particle concentrations decreases with increasing air speeds.

$N$ are calculated using an equation provided by Kleine (2019). An equation for the median volume diameter (*MVD)* and the *LWC* is offered in Droplet Measurement Technologies (2014) and the ice water content (*IWC)* is calculated during processing based on the work of Brown and Francis (1995) and Droplet Measurement Technologies (2013).

An analysis of the aforementioned measurement biases helps to develop a deep understanding of the probes' operation principles and improves the quality and accuracy of the results. Hence in the evaluation, no artefacts such as unnatural





multimodal shapes in the particle size distributions were found that could not be explained by means of natural processes and conditions in the atmosphere.

## 2.4     Characterization of mixed-phase clouds in convective systems over southern Germany

The meteorological situation over Western Europe on 8 July 2021 was dominated by a large amplitude trough from Scandinavia to Corsica. The trough induced a low-pressure system on its front side, as well as a large area of instability.
Together with the forced orographic lifting of the meridional low-level flow, strong vertical motions occurred which supported the growth of convective systems over the Alpine regions. These systems featured rapid supersaturation ($RH > 100\ \%$) conditions and a thick cloud cover, similar to tropical convective systems. Figure 4 features an equidistant Red-Green-Blue (RGB) satellite image displaying deep convection as well as the aircraft position at 17:50 UTC (blue) and shortly before (red). While satellite images provide a valuable overview of the large-scale meteorological situation, they did not display systems in

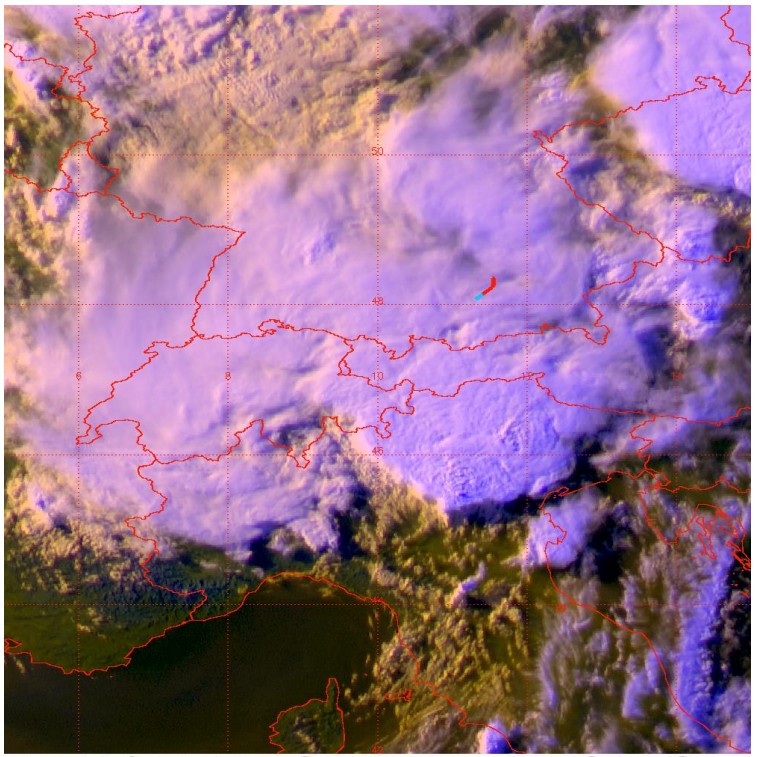

**Fig. 4: Equidistant Red-Green-Blue (RGB) image from the EUMETSAT-DLR MET-10 satellite on 8 July 2021 at 17:50 UTC depicting the deep convective system above the Alpine region and Northern Italy. Image was created and evaluated by DLR.**

lower levels beneath the thick cloud tops. Therefore, ground-based MOHp C-Band radar reflectivity cross sections obtained from range height indicator (RHI) scans were analyzed additionally to gain a more comprehensive overview. A total of 32 cross sections displaying the measured reflectivity in a two-minute interval are combined in a video and provided as supplement (Papke Chica et al., 2022). These, however, were merely used as reference as the aim of this work was not to provide an in-depth radar intercomparison. Nevertheless, such a comparison might be subject to future studies as simultaneous

in-situ and ground based radar station measurements of the same meteorological systems are not encountered very often. Fig. 5 displays a radar cross section at 17:51:16 UTC plotted over the latitude and displayed for altitudes up to 10000 m. The HALO flight path was marked in gray and allowed to determine whether the aircraft probed clouds of one or multiple meteorological systems.

Prior to evaluation, the flight section of interest was divided into five horizontal flight layers, with quasi isothermal conditions,

i.e., constant temperatures, equivalent to the aforementioned flight legs. At these layers, HALO remained at nearly constant altitudes and temperatures, as can be seen in Fig. 6. In the first layer (8873 m), $T$ remained at -35 °C with

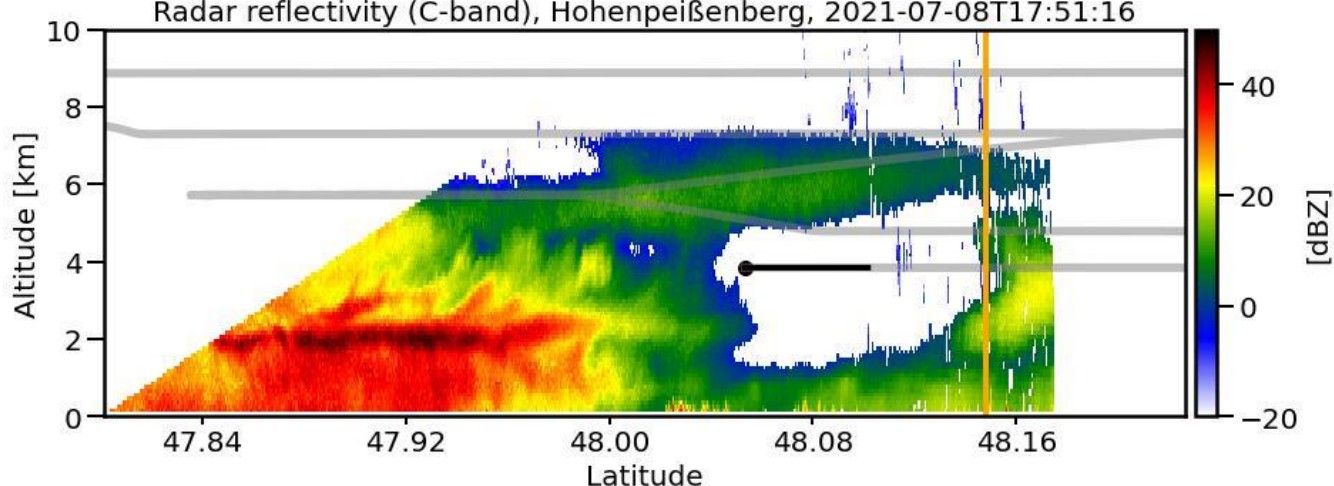

**Fig. 5: Radar reflectivity in dBZ obtained from range height indicator scans by the German weather service (DWD) at the climate reference station Hohenpeissenberg Meteorological Observatory (MOHp). Reflectivities are plotted over latitude and for altitudes between 0 km and 10 km in a two-minute interval. The orange vertical line marks the LMU radar station that was overpassed during this flight as well. The gray line marks the HALO flight path and the black line HALO's position one minute before the cross section is depicted. The full-length animation for the time frame 16:59:16 UTC to 18:01:20 UTC is available as supplement (Papke Chica et al., 2022).**

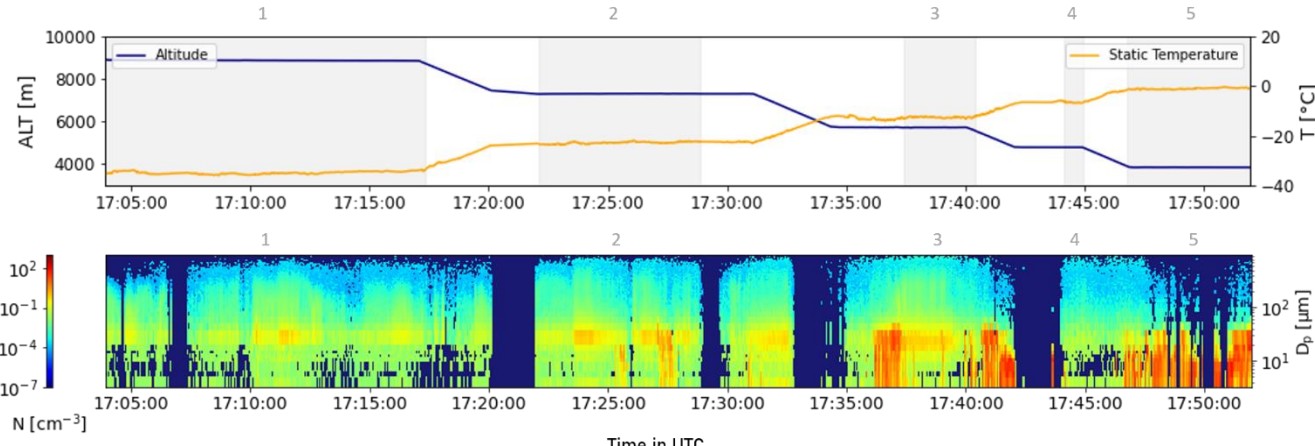

**Fig. 6: Upper panel: Altitude (blue) and temperature (orange) plotter over time showing the five evaluated flight layers (shaded, numbered gray) during the descent over Germany. Data were recorded with the Basic HALO Measurement And Sensor System (BAHAMAS). Lower panel: Particle size distributions versus time. The number concentration N per size bin is color coded. Data were obtained with the Cloud Imaging Probe (CIPg) and the Cloud Aerosol and Precipitation Spectrometer (CAS-DPOL).**

only minor deviations (± 1 °C), at the second layer (7303 m), $T$ remained around -23 °C, at the third (5707 m) at -13 °C, at the fourth (4783 m) at -7 °C and finally, at the last layer (3840 m) at -1 °C. The division was introduced to ensure constant ambient conditions when deriving the cloud microphysical properties. The exact mean temperatures and altitudes for each leg were

obtained from the Basic HALO Measurement And Sensor System (BAHAMAS) (Krautstrunk and Giez, 2012). Since MPCs mainly appear in a temperature range of -38 °C < $T$ < 0 °C (Lohmann et al., 2016b; Korolev et al., 2003), the data recorded over the course of the descent at $T$ between -36 °C and 0 °C provide an ideal overview of the cloud composition stages appearing in the mixed-phase temperature regime.

In addition to selecting constant ambient conditions, it was important to define at what times the aircraft passed through a

cloud. For this purpose, a cloud-threshold was implemented for all particles with $D_p$ > 3 µm at a $LWC$ > 0.01 gm⁻³, which is in agreement with values chosen in literature (Korolev et al., 2003). Additionally, a video composed of images from a GoPro installed in the aircraft by the University of Leipzig helped to determine at which parts of the flight HALO passed through clouds. An example of an image featured in the video is displayed in Fig. 3. It also gave an insight into the cloud opacity and at what times icing occurred on the probe housing. Ice started accumulating on the instruments' surfaces in layer 2, growing

larger at layer 3 and then remained until the descent to layer 5. The accumulation of ice reached its peak at layer 3.

The lower panel of Fig. 6 displays the particle size distribution as well as the total number concentrations $N$ (aspherical and spherical particles) over time for each layer. Note that the length of each flight layer differs, depending on the duration the plane remained in a cloud. Layer 5, at -1 °C, features the highest number concentrations exceeding 200 cm⁻³, but mainly contains particles with diameters between 3 µm and 40 µm, hence mainly in the CAS-DPOL size range. The first three layers,

at -35 °C, -23 °C, and -13 °C, depict similarly high $N$ up to 10 cm⁻³ with occasional concentration peaks (orange) that correlate with high updrafts (compare to Fig. 10) and higher concentrations of particles with diameters between 20 µm and 40 µm. The





PSD also depicts the highest concentrations of large particles with $D_p < 50$ µm. Layer 4 at -7 °C contains the lowest $N$ (1 cm$^{-3}$) and at the same time is the shortest cloud pass out of the five. Moreover, the layers strongly differ in $N$ ranges. While layers 1, 2 and 3 only contain a moderate number of particles ($10^{-4}$ cm$^{-3} < N < 10$ cm$^{-3}$), leg 5 exceeds $N$ of 200 cm$^{-3}$. This indicates

much denser clouds in the lower layers and optically thin clouds in the upper layers.

Additional differences and similarities of the cloud compositions in the particular layers can clearly be seen when taking a look at Fig. 7. It depicts the vertically resolved distribution of $N$ for spherical particles with $D_p < 50$ µm (CAS-DPOL and CIPg) and for aspherical particles with $D_p > 50$ µm (CIPg). The flight layers are numbered in gray and plotted with respect to latitude; Hohenpeissenberg being situated around 47.8 °N. Layers 1 to 3 show similarly high concentrations for ice particles

(0.1 cm$^{-3}$) and low concentrations for spherical particles (0.001 cm$^{-3}$ to 3 cm$^{-3}$). Even though these legs were probed in different altitudes, it appears as if they are part of the same convective cloud system. However, radar reflectivity cross sections (Fig. 5) suggest that during the first two legs at 8873 m and 7303 m the aircraft probed outflows of the convective system displayed in Fig. 4, then at 5707 m the aircraft passed the rear outflow of a storm cell that newly formed above the alpine region.

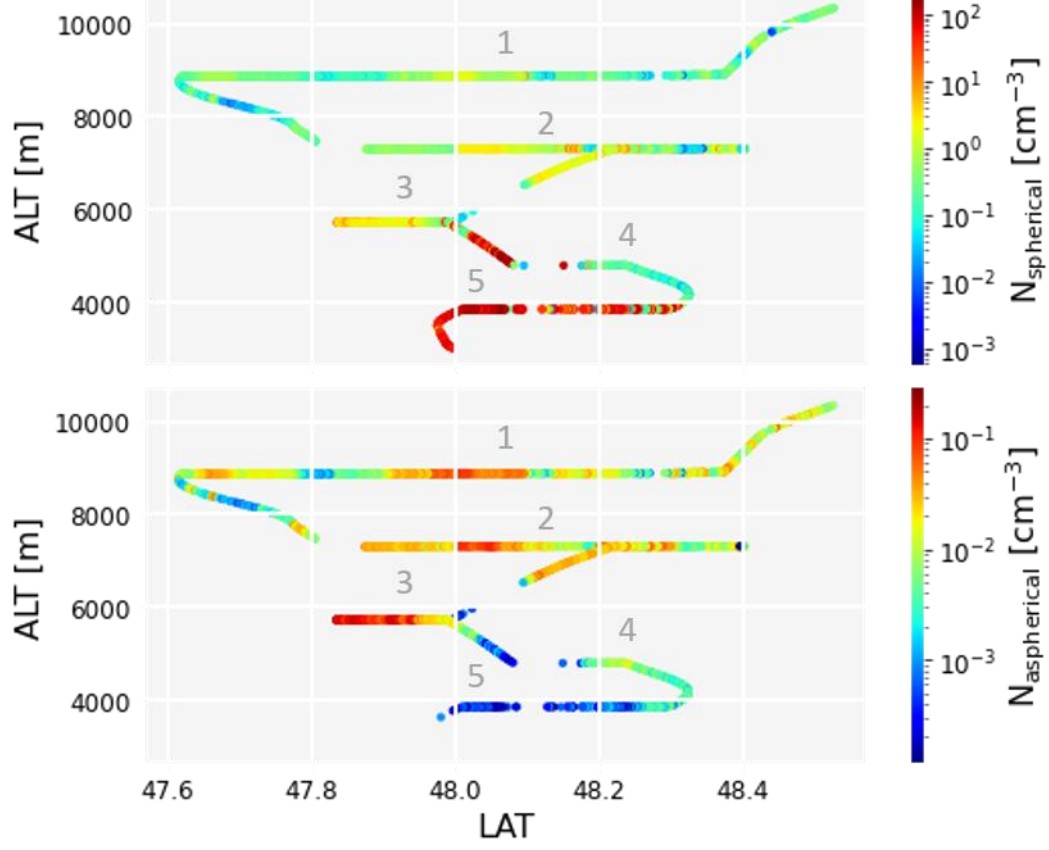

**Fig. 7: Graphs of the five flight layers (numbered) described in Sect. 2.4 for the time frame 17:00 - 17:55 UTC. The altitude (ALT) is plotted against the latitude (LAT). The color bars display the distribution of the $N$ for spherical particles with diameters between 3 µm and 50 µm measured by the CAS-DPOL and the CIPg (upper panel) and for aspherical particles with diameters > 50 µm measured by the CIPg (lower panel).**





Layer 5, however, seems to stand separately and shows high concentrations of spherical particles (200 cm$^{-3}$) and almost no ice

particles ($N < 0.0004$ cm$^{-3}$). Layer 4, again, differs from the above, with the lowest $N$ of spherical particles ($< 1$ cm$^{-3}$) and

moderate $N$ of aspherical particles (0.001 cm$^{-3}$). Here, radar reflectivities imply that at 4783 m HALO crossed the remaining

parts of the convective system probed in the first two legs, however it is also possible that this cloud "collided" and mixed

with the outflow of the storm cell probed in leg 3. In layer 5 (3840 m) newly formed liquid clouds not visible in the radar cross

section, due to low reflectivities of small particles, mixed with remainders of the old convective system were probed.

In Fig. 8, $N$ is subdivided into $N_{small}$ for particles with diameters $< 50$ µm and $N_{large}$ for particles with diameters $> 50$ µm, which

is equivalent to the phase differentiation threshold, however, $N_{large}$ can still contain large spherical liquid droplets. $N_{small}$ and

$N_{large}$ are graphically depicted in boxplots including 10 %, 25 %, 50 %, 75 % and 90 % percentiles, as well as mean and medium

values. Layer 1, at the lowest temperature, contains low concentrations of small particles with a median value at 0.3 cm$^{-3}$ and

with increasing temperatures at legs 2 and 3 $N_{small}$ increases to values with medians at 1 cm$^{-3}$ and 3 cm$^{-3}$.

**Fig. 8: Boxplots of calculated microphysical parameters measured by the CAS-DPOL and CIPg. Top: Number concentration s of particles with $D_p < 50$ µm ($N_{small}$) and $D_p > 50$ µm ($N_{large}$). Note that $N_{large}$ can still contain large spherical liquid droplets. Center: Median volume diameters $MVD$s for spherical (red) and aspherical (teal) hydrometeors. Bottom: Liquid (orange) and ice water content (blue).**





Unexpectedly, at $T = -7$ °C $N_{small}$ is much lower than in layer 1 (0.3 cm$^{-3}$). As expected, the highest concentration of particles

with $D_p < 50$ μm is found in the warmest layer at T = -1 °C with an arithmetic mean of 80 cm$^{-3}$. At the same time, this layer

contains the lowest concentration of particles with $D_p > 50$ μm; the largest amount is found in layer 3 at -13 °C. *MVDs* of

spherical and aspherical particles steadily increase up to layer 4 (-7 °C), which contains liquid or frozen droplets, as well as

ice with the largest *MVDs*. At the lowest altitude, droplets with the smallest *MVDs* and *MVDs* of aspherical particles with a

high degree of dispersion between 100 μm and 600 μm were probed. Values of *LWCs* and *IWCs*, which depend on *MVDs* as

well as on *N* are shown in the two bottom sets of boxplots of Fig. 8. The highest values are measured at -13°C, where particles

with the largest *MVDs* and relatively high *Ns* are found. Respectively at -1 °C, the lowest *LWCs* and *IWCs* are found, which is

due to small *MVDs* of spherical particles and low ice concentrations.

MPCs occur in meta-stable environments where supercooled droplets coexist with frozen droplets and/or ice crystals. How

fast a stable environment, i.e., glaciation, can be reached is determined by various parameters of which $RH_w$, $RH_i$, and $W$ play

an important role. During this flight, subsaturation with respect to water is mostly maintained for layers 1 to 4 (Fig. 9), while

at layer 5 the atmosphere is saturated with respect to water $RH_w = 100$ %. In this cloud layer, the air remains at an equilibrium

with the supercooled droplets and the atmosphere is supersaturated with respect to ice ($RH_i > 100$ %). A supersaturation with

respect to ice mostly prevails during the whole flight section, merely legs 1 and 4 show some longer deviations where both,

$RH_i$ and $RH_w$, drop under the saturation line and likely cause the cloud to partially evaporate. Whether supersaturation can be

maintained is determined by $W$. Supersaturation, in return, determines what microphysical processes can take place and as a

result, what particles exist and grow within the cloud. Therefore, the intensity of up- or downdrafts proves to be one of the

most important parameters determining convective cloud parameters. The highest updraft velocities are recorded by

BAHAMAS in the first three legs (Fig. 10). Especially at -23 °C and -13 °C a wide range of $W$ values are documented, reaching

velocities as high as ±3 ms$^{-1}$ and ±4 ms$^{-1}$. This wide distribution indicates strong cloud dynamics and a turbulent environment.

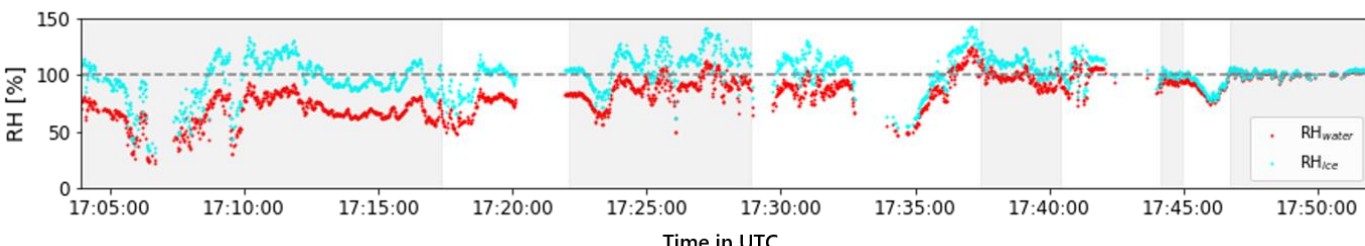

**Fig. 9: Relative humidity with respect to water (RH$_{water}$) and ice (RH$_{ice}$) plotted over time in UTC during the flight section above Southern Germany. Saturation at RH = 100 % is marked by a gray dashed line. While data for RH$_{water}$ were provided by the BAHAMAS instruments, RH$_{ice}$ was calculated using the saturation vapor pressures obtained by the BAHAMAS.**





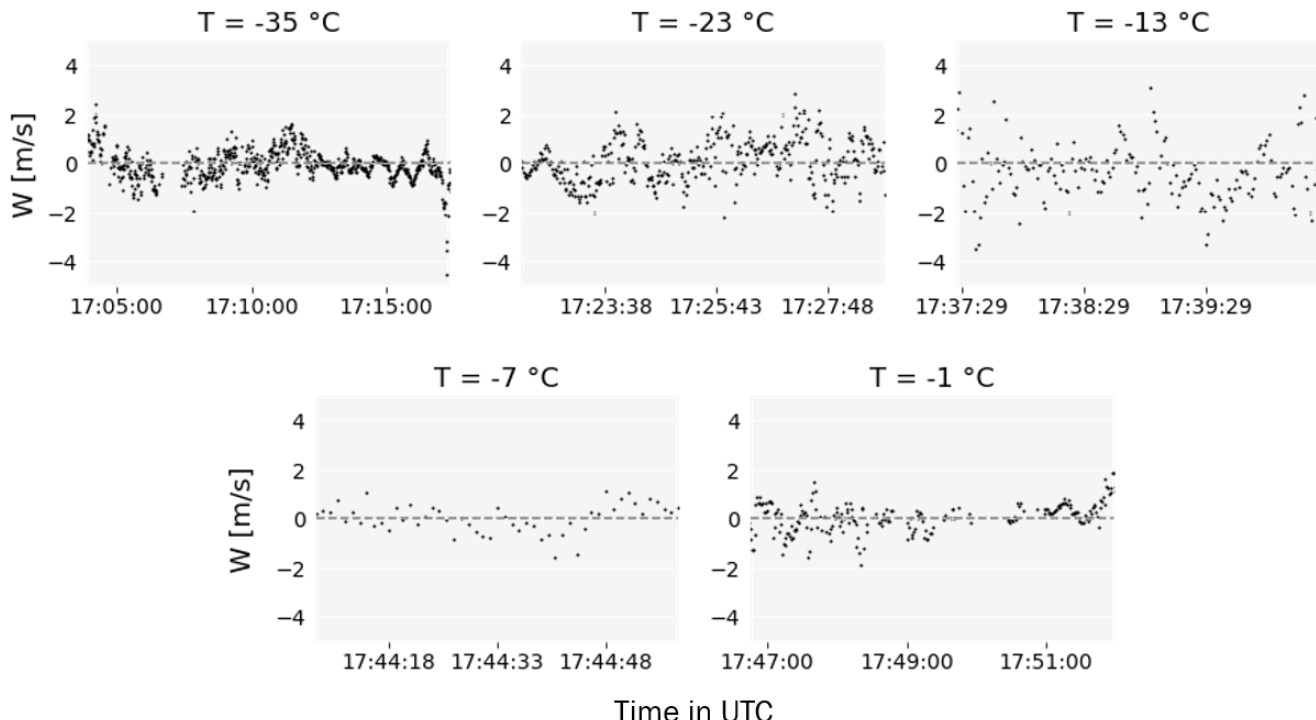

**Fig. 10: Vertical velocities (*W*) plotted over time for each second of the five flight layers evaluated during the flight section above Southern Germany with the HALO aircraft. The high vertical velocities varying between ± 4 ms$^{-1}$ strongly influence the cloud microstructure and emphasize the turbulent environment of this mid-latitude convective system. Data were obtained from the meteorological sensor system on HALO (BAHAMAS).**

High velocities, as found here, are more common in tropical convective clouds, whereas in mid-latitudes *W* between ±1 ms$^{-1}$ and ±2 ms$^{-1}$ appear more often. While in mid-latitudes updrafts of 1 ms$^{-1}$ are considered moderate, updrafts of 2 ms$^{-1}$ are already considered strong (Costa et al., 2017). At the cloud layers with -7 °C and -1 °C, *W*s vary between -2 ms$^{-1}$ and +2 ms$^{-1}$, which are more common for clouds in the probed latitudes.

In the following section, the described microphysical properties and ambient conditions within the cloud will be evaluated considering differences and similarities between the flight layers and possible phase determining ice nucleation and growth processes.

## 2.5    Classification of cloud layers

### 2.5.1    Classification of cloud layers 1, 2, 3 (-35, -23, -13 °C)

The cloud layers are sorted according to their particle size distribution type (Type 1/2), first, and then examined individually. An example of a Type 2 or *Large Ice* cloud can be found in flight layer 3. The PSD in Fig. 11 displays a clear second mass





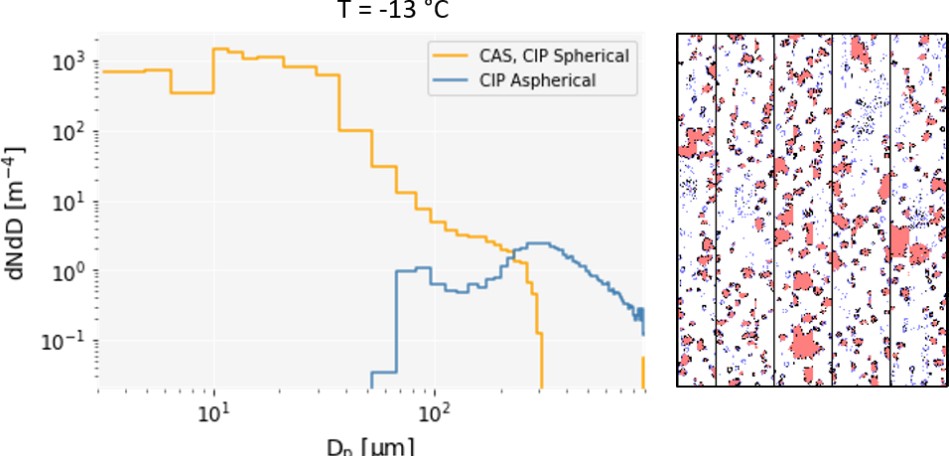

**Fig. 11: Left: Characteristic particle size distribution (PSD) of a *Large Ice* cloud at -13 °C. Particles are differentiated between spherical (orange) and aspherical (blue) shapes for particles larger than 50 μm. Spherical particles are either liquid droplets or frozen spherical particles and aspherical particles represent ice crystals. The evaluated data were obtained with the CIPg and CAS-DPOL in cloud layer 3 (Fig. 6) between 17:37:27 and 17:28:52 UTC. Right: Images of cloud particles taken by the CIPg. The distance between the vertical lines represent 960 μm.**

mode for aspherical particles with $D_p > 100$ μm (blue), while smaller spherical particles still dominate (orange). Further, Costa et al. (2017) describe that in *Large Ice* clouds $N_{small}$ typically lie below 1 cm$^{-3}$ and high concentrations of large ice crystals are present. While latter clearly applies for the concentrations found in cloud layer 3, $N_{small}$ still remains > 1 cm$^{-3}$ at -23 °C

and -13 °C (Fig. 8). Since shattering in the CIPg and CAS-DPOL is eliminated in layer 1 and 2 and not clearly registered in the CAS-DPOL at layer 3, it can be assumed that the cloud is not yet fully glaciated in layer 2 and 3 and still contains supercooled droplets. The existence of possibly liquid cloud droplets in *Large Ice* clouds at the given temperatures is not specified in Costa (2018) for mid-latitude cumulus clouds. Yet, this phenomenon is described for convective clouds in the tropics with higher $W$ similar to those probed during the evaluated flight section. It is also possible that this cloud was probed

in an intermittent stage between a *Large Ice* cloud and a *Coexistence* cloud, since it fits into the transition region between the two cloud types described by Costa (2018) for temperatures between -10 °C and -20 °C. However, *Coexistence* clouds maintain $RH_w > 100$ %, which is only found in roughly 5 % of the flight leg (Fig. 14) and the characteristic Type 1 PSD is not found in this layer.

The WBF process is assumed to take place at temperatures below -10 °C and at moderate $W$, predominantly at $W < 1$ ms$^{-1}$

(Korolev, 2007b). If updrafts exceed 2 ms$^{-1}$ it is assumed that the environment remains supersaturated with respect to water and ice crystals cannot form at the expense of water droplets (Korolev, 2007b). Even though in this flight layer $RH_w$ mostly remains just below saturation ($RH = 100$ %), supersaturated with respect to ice, and the temperature at -13 °C, the environment might have been too turbulent for the WBF process to fully develop and glaciate the cloud. It is also very likely that due to high updrafts, small particles might have been lifted up from lower cloud layers. At some parts, the ambient conditions are





subsaturated with respect to water and ice, which correlates with downdrafts (compare Fig. 9 and Fig. 10) and cause the cloud
to evaporate.

In addition to evaluating the PSD, a closer look is taken at the particle images captured by the CIPg (Fig. 11). Here, large ice
aggregates with diameters up to 900 μm prevail, and smaller crystals and possibly large frozen or supercooled droplets can be
observed. Medium sized ice crystals might have initially formed through depositional growth in the upper cloud layers until

they reached terminal velocities high enough to fall against the updraft and sediment into the probed layer. On their way down,
particles grew by accretion when colliding with supercooled droplets. This growth, referred to as riming, explains the large
particle sizes seen in Fig. 11. Growth by aggregation is unlikely in this temperature range since the quasi-liquid layer (QLL)
that forms on particles' surfaces and generates a sticking effect is too thin at T < -10 °C (Lohmann et al., 2016b). The high
concentration of small particles likely also results from ice multiplication through fracturing, when graupel encounters slower

falling dendritic crystals (Lohmann et al., 2016b), which are common in a temperature range between -10 °C and -20 °C.
Additionally, lightning was observable during the flight which also led to shortening of the flight layer and exiting the cloud
earlier for the sake of flight safety. In cumulus clouds, the main charge region lies between -10 °C and -20 °C and can have its
origin in a charge generation between ice crystals moving up, due to high $W$ and graupel moving down (Lohmann et al.,
2016b). Both are given in this cloud layer. Due to large concentrations of ice crystals, these clouds appear thinner with different

radiative properties than denser Type 1 clouds.

Layers 1 (-35 °C) and 2 (-23 °C) show similar particle size distributions as well as particle images to Layer 3, which is why
they were categorized as *Large Ice* clouds as well. Especially the coldest leg fits this classification, as it contains the lowest
concentration of small particles ($N_{small}$ < 1 cm$^{-3}$) out of the three legs and more moderate $W$, also favoring the WBF process.
Based on an aspherical fraction analysis conducted by Costa (2018), it is very likely that all particles between 20 μm and 50

μm are ice particles in Type 2 clouds. Due to a temperature of -35 °C it is also possible that the remaining particles froze
homogeneously in layer 1, therefore it can be assumed that this cloud was completely glaciated.

At some sections within layer 2 supersaturation with respect to water and ice can be observed (Fig. 9) and an analysis of the
PSDs reveal that *Coexistence* and *Secondary Ice* clouds were also present. Additionally, it is very likely that all three upper
legs contain secondary ice particles due to high vertical up- and downdrafts.

Within one cloud multiple processes are found which change on irregular time scales. Therefore, it can be assumed that
microphysical growth processes variate within a cloud layer and strongly influence particle concentrations, making it difficult
to classify a cloud only based on the PSD.



### 2.5.2 Classification of cloud layer 5 (-1 °C)

A clear example of a *Mostly Liquid* cloud is presented in flight layer 5. Here, very high $N$ exceeding 100 cm$^{-3}$ were probed,
mainly featuring spherical particles with $D_p < 50$ μm. The main mode of the particle size distribution is located in the small
particle size range (orange), which is characteristic for such clouds (Fig. 12). While ice is still present in low concentrations,
$N_{small}$ by far dominates. Especially at the end of this layer, only small amounts of ice crystals are found, which can be seen in
the particle image of CIPg. In spite of being sparse, ice is present in a wide variation of sizes, reaching from *MVD*s of 100 μm
to 600 μm (Fig. 8). In this layer, large ice crystals sedimented from upper cloud layers or grew by aggregation, which results
from ice crystals colliding and clumping together at the QLL. These aggregates are characterized by their highly irregular
shapes and sizes and can be observed in the CIPg images. Additionally, riming occurred when supercooled droplets froze upon
contact with ice particles. Another process that most likely took place in this supercooled environment is contact freezing,
resulting from an ice nucleating particle (INP) colliding with a droplet and subsequently freezing. In subsaturated conditions,
as found here, a special form of contact freezing appears, referred to as "contact nucleation inside out" (Durant and Shaw,
2005). In this case, evaporating, i.e. shrinking supercooled droplets already contain an INP and freeze upon contact with its

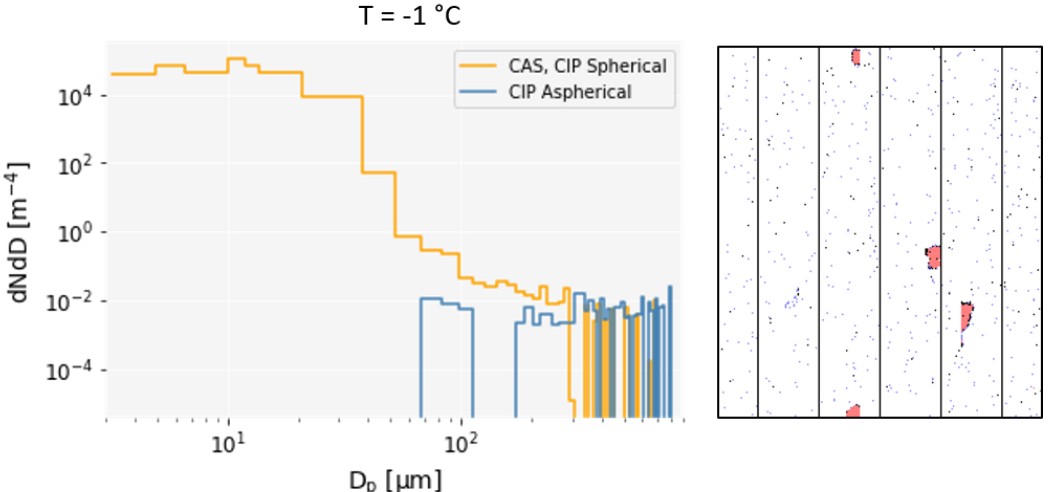

**Fig. 12: Left: Characteristic PSD of a *Mostly Liquid* cloud at -1 °C. Particles are differentiated between spherical (orange) and aspherical (blue) shapes for particles larger than 50 μm. Spherical particles are either liquid droplets or frozen spherical particles and aspherical particles represent ice crystals. The evaluated data were obtained with the CIPg and CAS-DPOL in cloud layer 5 (Fig. 6) between 17:46:46 and 17:51:58 UTC. Right: Cloud particles imaged by the CIPg. The distance between the vertical lines represent 960 μm.**





surface (the air-water interface). Moderate $W$ ($\approx 1$ ms$^{-1}$) and the $RH_w$ mainly staying close to the saturation line in this leg also support the classification as *Mostly Liquid*.

An analysis conducted by Costa et al. (2017) evaluating the aspherical fraction (AF) of particles with diameters between 20 µm and 50 µm in *Mostly Liquid* clouds, reveals that 0 % are frozen. Since the same instrument (CAS-DPOL) was used for the AF

analysis and the ambient conditions were similar it is assumed that in this layer particles with $D_p < 50$ µm are also liquid. The aforementioned processes leading to the growth of a few large ice particles explain the composition of hydrometeors seen in the CIPg image. *Mostly Liquid* clouds are thought to be young clouds after droplet condensational growth (Costa et al., 2017) and are common in mid-latitudes.

### 2.5.3 Classification of cloud layer 4 (-7 °C)

The next layer's PSD differs from the other two by neither displaying a distinct mode in the small, nor in the large particle size range (Fig. 13). Therefore, it was not possible to directly place it into one of the two main cloud groups (Type 1/2). The curtain plot of layer 4 at -7 °C (lower panel Fig. 6) presents very few to no particles between 5 µm and 10 µm and low concentrations (< 1 cm$^{-3}$) of particle sizes up to 5 µm and from 10 µm to 50 µm. Due to moderate to high $W$ between 1 ms$^{-1}$ and 2 ms$^{-1}$, it is possible that a significant amount of ice active INPs were carried into the cloud. At the given temperature, these INPs likely

are of biological origin, since mineral dust INPs are more likely to initiate freezing at temperatures below -20 °C (Kanji et al., 2017; Augustin-Bauditz et al., 2014). The existence of these particles combined with subsaturated conditions with respect to water and mostly saturated conditions with respect to ice likely initiated heterogeneous ice nucleation via condensation freezing

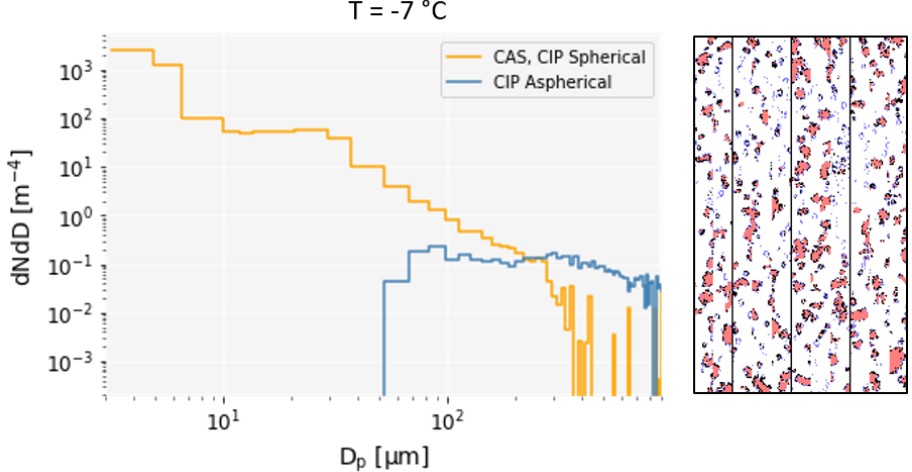

**Fig. 13: Left: Particle Size Distribution of a cloud layer measured at -7 °C transitioning from a *Secondary Ice* to a *Large Ice* cloud. Particles are differentiated between spherical (orange) and aspherical (blue) shapes for particles larger than 50 µm. Spherical particles are either liquid droplets or frozen spherical particles and aspherical particles represent ice crystals. The evaluated data were obtained with the CIPg and CAS-DPOL in cloud layer 4 (Fig. 6) between 17:44:08 and 17:44:58 UTC. Right: Cloud particles imaged by the CIPg. The distance between the vertical lines represent 960 µm.**


or "contact nucleation inside out". Condensation freezing starts at water subsaturated conditions with an air parcel containing an INP and once water saturation is approached ice seeds can form on the INPs surface (Lohmann et al., 2016b). Since water

saturation was barely reached and water subsaturated conditions caused cloud droplets to evaporate, it is more likely for "contact nucleation inside out" than for condensation freezing to have occurred in this layer. This type of freezing is more likely to happen at higher $W$ ($\pm 2$ ms$^{-1}$), which are given here as well. In addition to this ice nucleating process, riming occurred in this cloud, thereby generating large irregular shaped ice crystals as seen in the CIPg images. Larger crystals present in this cloud also sedimented from upper cloud layers. A process in this layer that produced small secondary ice in shape of "splinters"

is the Hallett-Mossop process (Hallett and Mossop, 1974). During this mechanism splinters are produced at temperatures between -3 °C and -8 °C, while requiring the presence of supercooled droplets with $D_\mathrm{p} > 35$ µm, as well as rimed ice. In this cloud layer, supercooled droplets and rimed ice crystals are present in various sizes, including diameters necessary for the Hallett-Mossop process to occur, as is displayed in the two middle boxplots of Fig. 8 and in the PSD. While at -7 °C the largest *MVD*s for spherical and aspherical hydrometeors out of the five layers are found, droplets with diameters between 3 µm and

50 µm are present in high concentrations (Fig. 13). At the given temperature cylindrical ice shapes like columns or needles are typically found. In this cloud with low $RH_\mathrm{w}$, hollow columns and solid prisms are more likely than needles. Due to the relatively low resolution of the CIPg compared to other particle imagers, these hollow columns cannot be seen as clearly in Fig. 13, nevertheless some hydrometeors are elongated which implies a cylindrical shape. Owing to partially high updrafts, a temperature favoring a QLL, a $RH_\mathrm{i}$ close to saturation, as well as $RH_\mathrm{w} < 100$ %, ice crystals might have collided with others,

stuck together and formed aggregates of columns. Such aggregates of columns are rare and even though it cannot be said with absolute certainty that such crystals are present in this cloud, the ideal conditions favoring their growth are given. Since Fig. 13 does feature particle aggregates, it seems reasonable for these to be composed of columns. The CIPg image in Fig. 13 resembles the CIPg images of Fig. 11, but differs by containing more medium and small sized ice crystals and appearing very dense. It is likely that as a result of the described processes, this layer's composition was dominated by the production of

secondary ice, typical for *Secondary Ice* clouds (Costa et al. 2017; Gayet et al. 2012) and might have been probed in an advanced cloud stage shortly before it was fully glaciated.

The typical PSD the authors describe for this cloud type, presenting a clear mode for particles with $D_\mathrm{p} < 50$ µm, is most likely not found due to the late stage of this cloud's life time, where a lot of particles in this size range either evaporated or settled as rime on ice crystals. The image of this cloud conveys a dense distribution of particles with high concentrations of small

fragments, which based on the backscatter analysis by Costa et al. (2017) are aspherical. Consequently, this *Secondary Ice* cloud differs from the other cloud layers in radiative properties.

## 2.6    Comparison to previous in situ cloud observations in the mixed-phase temperature regime

A statement about the probability of cloud occurrences in mid-latitudes as conducted for a total of 38.6 h of data in Costa et al. (2017), cannot be made at this point due to lower statistics. Nevertheless, the long flight layers inside clouds during this





case study are used to derive a rough estimate of the occurrence of different cloud types within a cloud (Fig. 14). For this purpose, cloud sections are classified based on the PSD type and the *RH*. While the PSD type helps to distinguish between a Type 1 and 2 cloud, the *RH* allows for a closer differentiation of the microphysical processes controlling the cloud formation and growth. As an example, the Wegener-Bergeron Findeisen Process predominantly occurs when $RH_w < 100\,\%$ and $RH_i > 100\,\%$, and the PSD displays an additional mode for aspherical particles with diameters greater than 50 µm (PSD Type 2). The

results of these estimates for each isothermal flight layer (- 35 °C (238 K), -23 °C (250 K), -13°C (260 K), -7 °C (266 K), -1 °C (272 K)) are visualized in Fig. 14.  Since we find different cloud types within some cloud layers, a coexisting variation of microphysical processes can be assumed. Based on the results of this evaluation, the most common cloud group is *Large Ice*, which is observed in three of the five flight layers. Since the WBF process is thought to be the most dominant process in mid-latitudes (Boucher et al., 2013), this result does not come as a surprise and is also in accordance with Costa et al., (2017).

The measurements in mid-latitudes of Costa (2018) were conducted above the UK and mainly focused on stratus and stratocumulus clouds in February and March, whereas the data in this work was collected in deep convective cumulus clouds above Southern Germany in July. An explanation for differing cloud types within the cloud layers of this convective system

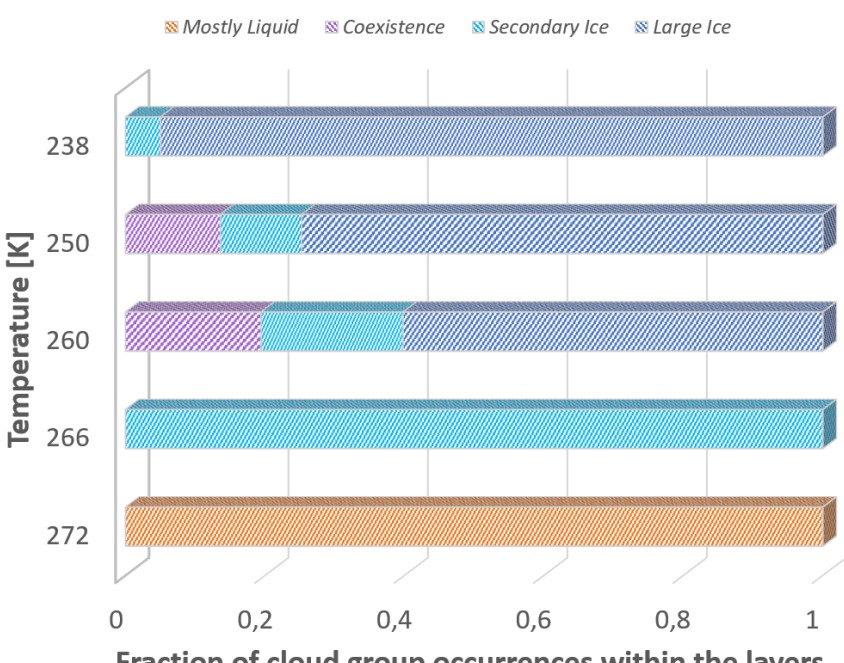

**Fig. 14: Cloud group occurrences within a probed flight layer during the CIRRUS-HL flight above Southern Germany on 8 July 2021 between 17:00 and 17:55 UTC. In total five flight layers at quasi constant temperatures (272, 266, 260, 250 and 238 K) were probed and classified into four groups (*Mostly Liquid*, *Coexistence*, *Secondary Ice*, *Large Ice*). The classification was based on ice nucleating and growth processes and microphysical cloud properties measured with the CAS-DPOL and CIPg. Within some of the flight layers multiple cloud groups were found, as depicted in this figure.**





compared to Costa's measurements in mid-latitudes are higher and more variable $W$. While Type 2 clouds dominate they cannot be classified as clearly, since $N$ for small particles mostly remain $> 1$ cm$^{-3}$. These $N_{small}$ indicate the existence of

supercooled cloud droplets and small ice particles with 20 μm $< D_p < 50$ μm. An explanation might be the more turbulent environment given in this flight that hinders the WBF process from fully developing. This presents a major difference to the results of Costa et al. (2017), where the existence of possibly liquid cloud droplets in Type 2 clouds was only observed in tropical latitudes. This emphasizes the importance of accounting for $W$ when classifying MPCs.

At higher temperatures between -10 °C and -20 °C, *Coexistence* clouds appear when high updrafts cause the $RH_w$ to remain

near to and slightly above 100 %. Costa (2018) refers to this temperature range as a transition region between *Coexistence* and *Large Ice* clouds. During this flight, *Coexistence* clouds are only found in small sections within clouds at -23 °C and -13 °C (Fig. 14). A classification of these sections based on the PSD expecting a clear mode for particle sizes $< 50$ μm, does not always reveal the anticipated Type 1. However, the requirement of supersaturation with respect to water and ice (correlating with higher $W$) is met suggesting that the clouds might have been probed at a transition between Type 1 and 2. This implies

the presence of a variety of ice nucleating and growth processes within a cloud layer, rather than just one. These strongly depend on $W$ and $RH$, thereby making it difficult to classify the overlapping process-based cloud type and composition solely based on the modes of the particle size distributions. This highly variable distribution and interaction of microphysical processes within a convective system and cloud layer seen here, is one of the major difficulties when it comes to quantifying MPCs in climate and cloud models, because they directly affect and cause variations of the cloud particle composition and

radiative properties. At even higher temperatures (255 K, 260 K, 265 K) Costa (2018) describes the appearance of *Secondary Ice* clouds that are generally less frequent in mid-latitudes than in the tropics and correlate with moderate to high updrafts. While again, a clear classification based on the PSD type is not possible for this cloud group in this work, it is still found at -7 °C based on other parameters (Sect. 2.5.3). When comparing the resulting occurrences of different cloud types, *Secondary Ice* clouds occur more often than previously found by Costa et al., (2017) in mid-latitudes.

The lowest layer with the highest temperature of the probed convective system was classified as *Mostly Liquid*, which is equivalent to the results of Costa (2018). Therefore, it can be assumed that the particle composition of the lowest layer of convective clouds is very similar to stratocumulus or stratus clouds.

To conclude this comparison, estimates of different process- and composition-based cloud type occurrences within the probed mid-latitude convective system in this case study lie between previous results for mid-latitudes and the tropics.

**3    Summary and outlook**

In situ observations of strong convective clouds in the mixed-phase temperature regime are rare and hence MPCs are likely to be misrepresented in climate and cloud models, often underestimating the cloud supercooled droplet fraction. In order to broaden the database of airborne in situ measurements in the mixed-phase temperature regime, mid-level convective clouds were probed above Southern Germany with the Cloud and Aerosol Spectrometer (CAS-DPOL) and the Cloud Imaging Probe



(CIPg). The meteorological situation over Western Europe on 8 July 2021 featured a thick cloud cover with deep convective cells almost comparable to tropical systems, with vertical velocities up to ± 4 ms$^{-1}$.

In this study, four process- and composition-based types of clouds with different liquid and ice fractions and hence differing radiative properties are found and classified according to Costa et al. (2017): *Secondary Ice*, *Coexistence*, *Mostly Liquid,* and *Large Ice*. Microphysical properties of particles between 3 µm and 960 µm were analyzed during descent along five different

stepwise flight layers, altitudes, and temperatures within the mixed-phase temperature range (-38 °C to 0 °C). Clouds enclosed in the first three flight layers (-35 °C, -23 °C, -13 °C) were mostly classified as Type 2 or *Large Ice*, since the majority of these layers' particle size distributions display a clear second mode for large aspherical particles with $D_\mathrm{p} > 50$ µm. Furthermore, particle images recorded by CIPg feature very large ice aggregates with diameters up to 900 µm. In layer 2 and 3 (-23 °C, -13 °C) higher concentrations of small particles ($N_\mathrm{small} > 1$ cm$^{-3}$) compared to the first layer (-35 °C) were found,

indicating that the WBF process was not fully developed due to high updraft velocities. Other explanations for high $N_\mathrm{small}$ ($D_\mathrm{p} < 50$ µm) include the presence of the *Coexistence* type within the cloud layer or fracturing, causing high concentrations of secondary ice. A *Mostly Liquid* cloud was classified in layer 5 (-1 °C) with $N$ exceeding 100 cm$^{-3}$ and a clear main mode for small particle diameters. Here, CIPg images display a large amount of supercooled liquid droplets and only few irregular shaped ice aggregates that precipitated from upper cloud layers. Additionally, sintering and riming contributed to the growth

of ice particles in this dense cloud. Cloud layer 4 at -7 °C differs from the other cloud types. The dense accumulation of small, medium, and large sized ice particles presented in this layer's particle image implies a *Secondary Ice* cloud, which was evoked by fracturing and the Hallett-Mossop process and thus the production of splinters. It is also possible that rare column aggregates formed, as the probed conditions were optimal for this type of ice crystal growth. To summarize, cloud layer 4 was probed in a late stage of *Secondary Ice*, where a lot of small particles already evaporated or settled as rime on large ice crystals.

A comparison to a previous study also demonstrates that, while the classification of *Mostly Liquid* clouds in low-level stratus, stratocumulus or cumulus clouds is fairly clear, a classification of the remaining composition-based cloud types in convective systems is difficult, due to strong up and downdrafts. The wide distribution of vertical velocities found in this system indicate strong cloud dynamics as are also found in tropical systems causing a strong fluctuation of relative humidities. As a result, a complex variation of temperature, relative humidity, and vertical velocity dependent processes initiating ice nucleation and

growth, as opposed to just one, determine the phase composition within the cloud layers. This study provides a comprehensive evaluation of underlying cloud mechanisms as well as a characterization of microphysical properties and the ice and liquid fractions within convective systems and suggests the WBF process might not be as prominent as assumed for MPCs in mid-latitudes.

Over the past years, there have been many contributions and advancements in the classification and phase identification of

MPCs. There are however, still many uncertainties associated with instrument-related measurement biases as well as the discrimination between liquid and ice particles. Especially the latter proves to be difficult as it assumes that ice particles are irregularly shaped, which excludes frozen droplets and heavily relies on the optical array probe size resolution. For further evaluations, it is therefore highly recommended to additionally analyze the backscatter signal of the CAS-DPOL to further





differentiate particle shapes in a size range between 10 μm and 50 μm. However, an uncertainty remains for the phase of
particles with sizes < 10 μm, which are assumed to be liquid droplets and the exact distinction, especially for particles sizes <
5 μm, remains an unresolved technical challenge.

The observations and results of this unique case study contribute to the database of in situ measurements in MPCs and
additionally provide insights into less frequently evaluated mixed-phases in convective clouds in mid-latitudes. Nevertheless,
it is important to obtain more statistics on microphysical processes and properties in this cloud species for mid-latitudes, as
this study was limited to a specific system and probed in unusually high updraft conditions. Additional comparisons to low
layer MPCs might also prove valuable, as phase-determining processes can diverge due to differences in vertical velocities
and relative humidities. The descent and parts of the cloud measurements were performed near the ground weather observation
stations of Hohenpeissenberg and Munich and thus invite for intercomparison with radar and lidar measurements. Further
evaluations and comparisons of in situ measurements of clouds in the mixed-phase temperature regime will help to improve
the understanding of cloud glaciating-mechanisms, hydrometeor sizes and shapes, as well as the phase distribution of liquid
and ice within different cloud species. This can enable a more accurate representation of MPCs and their radiative properties
in cloud and climate models.

*Data availability*

Data are available at the HALO data base at https://halo-db.pa.op.dlr.de/.

*Competing interests*

Some authors are members of the editorial board of journal Atmospheric Chemistry and Physics. The peer-review process was
guided by an independent editor, and the authors have also no other competing interests to declare.


*Acknowledgements*

We thank the DLR flight crews for excellent flight operations in the strong convection eventually encountering icing
conditions. Thank you as well to the Leipzig Institute of Meteorology team for providing the GoPro images of the flight. This
work was supported by the DLR Aeronautics Research Programme, and by the German Science Foundation within SPP-1294
HALO (contract no VO1504/6-1 and VO1504/7-1) as well as the CRC TP-Change 301. M.P and C.V. were supported by the
Helmholtz excellence programme (grant number W2/W3-060).

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
