# Peer review of "Process-based microphysical characterization of a strong mid-latitude convective system using aircraft in situ cloud measurements"

_Atmospheric Chemistry and Physics, 2022_

## Referee Comment (RC1)

I'm underwhelmed with this article. I'm not sure that there's any new physics or knowledge gained. What would have been interesting is the authors directly related the particle size distributions to the updraft velocity. Likewise, relating the vertical velocity to the relative humidity would have been interesting and a validation of the relative humidity measurements. There is no discussion of the measurements of the relative humidity, and how they related to the interpreted spherical and non-spherical particles. No particle images are shown, so that one cannot draw interpretations about the presence of secondary ice particles. Another point is the misinterpretation of the vertical velocities in mid-latitude convective clouds. There is a wide body of information out there which shows that vertical velocities of 20 m/s or higher are present in mid-latitude convective clouds and that 4 m/s is about what is observed at cloud base in convective clouds. The literature cited is mostly recent, not drawing upon measurements and interpretations from earlier studies. No direct measurements of the liquid water content is presented. This would have been very valuable to validate the interpreted particle types. Likewise, measurements at temperatures warmer than 2 or $3^0C$ would have been useful to evaluate the particle habits. No direct measurements of the condensed water content are available, unfortunately.

My specific comments appear below.

Line 28: Coexistence > Mixed Phase

30 midlatitude stratiform clouds

33: precipitating large ice

55: microphysical

55: and in altocumulus

80: I totally disagree with +/- 4 m/s being exceptionally high for midlatitude convective clouds

94: 50 microns is not "Large Ice". I would just call this ice phase. Large ice to me is >0.5 cm. Unfortunately, no measurements of particles above about 1 cm, and with a probe with a large sample volume, is available from the study.

110 "at lower temperatures"  Not specific and incorrect

Section 2.3 is a very good discussion of potential errors and how they have been treated. Nonetheless, I'm still uneasy about the removal of shattered particles, particularly for the CIPg in the small size channels

Section 2.3. How was the relative humidity measured? What is the accuracy.

212. sample volume,

216 In the case

234. Interesting that the pinhole diameter was decreased to reduce coincidence effects

245: artifacts

302 I wouldn't consider large to be 50 microns

346-347. Not correct for mid-latitude convective storms.

360 "eliminated" to "reduced"

Important: It would have been nice to have a sampling leg at temperatures above 0C to check your habit classification schemes. Also, a direct measurement of the liquid water content with a King type probe

406. Is it possible that the low concentrations of ice crystals is due to misclassification in your habit identification scheme

407: "amounts" to "concentrations"

506. These are not "strong convective clouds" for midlatitudes. See for example https://doi.org/10.1175/JAMC-D-12-0185.1

516: Large ice.

524: sintering. How do you know that growth was occurring through sintering? Do you mean aggregation?
Sentence beginning on line 525. Without particle images shown, I'm uncomfortable with your statement about the implication of the presence of secondary ice

---

## Referee Comment (RC2)

**Overview:**

This paper utilizes in situ aircraft and ground-based remote sensing observations of a midlatitude convective cloud system to describe the microphysical distributions as a function of temperature. The authors employ microphysical insights into characteristic hydrometeor types and growth regimes by combining measurement with state parameters, mainly temperature and RH. While this study provides a thorough evaluation of the case at hand, it has several flaws, some of which appear to originate at a fundamental level.

The Type 1 & 2 classifications seem arbitrary and unimportant, and the phase classifications skeptical. What was the point of using this classification as opposed to just doing a case study of the observed event and describing the thermodynamic and microphysical processes that are potentially active at each flight leg/temperature level? Moreover, the description of glaciation here really just matches the general conceptual model of deep convective mesoscale cloud systems and the dependence of glaciation on temperature. In addition, the authors repeatedly make the statement that tropical convective updrafts are stronger than typical midlatitude continental updrafts, which is untrue, especially when considering organized MCSs or supercells. It is true that in situ observations are rare for continental convection, but retrieval methods (e.g., Dual-Doppler analysis) have been performed.

This manuscript's potential lies in the potential ability to describe rare observations of a single deep convective cloud system. It would make more sense to me to describe the event and the vertical structure of the system generally, including inferences based on microphysical characteristics like the PSD/particle images and the thermodynamics. Moreover, the classification just seems arbitrary. One would expect for "Mostly liquid" clouds to exist near the melting level and transition to fully glaciated at colder temperatures. A classification algorithm is not needed to determine this. The structure of the manuscript also needs some work, as described below.

**General comments:**

- There is a lot of information and results being presented in the Introduction and it makes things a bit exhausting, especially when you're referencing ahead to figures and sections much later in the paper. I'd recommend merging some of this information with the Results in general and keeping the introduction limited to precursor information and a brief experiment outline.
- You're going to have to convince me that a 50 micron threshold is sufficient for determining that the larger mode is ice. Since we don't have descriptions of the instrumentation yet (in the introduction), we have no idea what's happening at the surface. Rain drops are much larger than 50 microns and can and do exist at subfreezing temperatures in convective systems through lofting. Knowing where that second (ice or liquid) larger mode often exists in size space would be an important description.
- Fig. 1 makes no sense when it is introduced. The figure seems a bit arbitrary, actually.
- Table 1: I think we need some context on why these threshold values are chosen and what they're based on. What are typical concentrations in midlatitude continental

clouds? You cite Costa et al. (2017) a lot, which I'm assuming describes this discrimination algorithm, but details need to be included here as well, even briefly.

- Section 2.2: You are using a lot of space to detail particulars of the algorithm (methods) and haven't really yet presented the algorithm formally, besides in the Introduction where it shouldn't belong. I'd recommend cutting down this Section to give hard specifics on the instrumentation and save the algorithm classification details for the next section

- I appreciate the authors' robust description of instrument uncertainty in Section 2.3. However, I believe this should belong in Section 2.2 when you introduce the instrumentation.

- I'd suggest putting Sections 2.1-2.3 in their own higher-order section, and starting with results (your Section 2.4 onward) in a separate section.

- You didn't say anything about the potential for particles with sizes > 940 microns—your upper instrumentation limit. Is it possible that large aggregates, lofted raindrops, or even graupel were present above this threshold? Radar reflectivity might help to elucidate this. Just a statement on this uncertainty would suffice.

- You clearly show that there are still small spherical particles at colder temperatures for layers designated as "Large Ice". Understanding that these may be liquid droplets or tiny ice fragments from SIP processes, isn't there still a l likelihood that these are mixed-phase clouds (i.e., your definition of Coexistence)? This ambiguity makes it difficult to understand the point of your classification scheme.

**Specific Comments:**

Abstract (Line 32): +/- 4 m/s is not abnormally strong of midlatitude continental deep convective updrafts

Line 54: "mycrophysical" should be "microphysical"

Line 56: Something went wrong here. The citation needs to be in parentheses and the following phrase "in altocumulus clouds" doesn't contextualize with the beginning of the sentence.

Line 69: should be "understanding *of* stratiform MPCs"

Line 83: See comment about abstract. These are not unusually strong at all

Line 91: We need some context of what these instruments measure. If you describe it later, just a short phrase will do.

Line 95: I think I see why you've labeled this "Large Ice", but it is going to be a bit confusing for the reader because 50 microns is rather small ice, especially in convective systems where graupel and snow are prevalent

Lines 151-152: need to break sentences before "second" and again before "finally"—this is a run-on sentence

Lines 153-157: What is the point in numbering arbitrary flight levels? I would remove these and just state the altitudes or give an altitude range

Line 157: Remove comma after "both"

Lines 174-175: Any evidence that most particles with Dp < 50 microns in "Large Ice" clouds are frozen? This seems fundamentally flawed because you're using a 50-micron threshold to distinguish been this class and the other 3.

Line 176: the description here of "Secondary Ice" is very confusing, because you haven't yet introduced how this distinction is made.

Line 186: What did you do to detect and negate out-of-focus images?

Lines 281-283: Several grammatical errors in this sentence owing to the separation of flight level descriptions. Recommend breaking this up into more sentences and/or fix the comma separation mistakes.

Line 296: Except they are *not* showing the total number concentrations, they are showing the number concentrations per size bin

Lines 301-302: I don't understand this statement. What do you mean by "large particles" if Dp < 50 microns?

Lines 309-310: this is a bit confusing without reference to expected concentrations for liquid and ice. You say ice is at a high concentration of 0.1 g/m3 but then state the liquid concentrations are low at an upper bound of 3 g/m3. Be sure to contextualize these values with what is considered high or low for each phase in this cloud regime

Lines 310-311: Aren't all of these clouds part of the same convective cloud system? I would clarify this and refer to parts of the cloud system that exist within the same cloud microphysical regime.

Lines 315-316: Again, context is needed, at least in the initial discussion, on what baseline values are for "high" and "low" in regard to phase.

Line 321: I think it is important to mention how prevalent large spherical drops are present. This system looks like it's raining. Perhaps give a frequency of how often spherical drops > 50 microns are present? Isn't that information available through the previously mentioned processing algorithms?

Line 325: Why is this unexpected? Because of what was shown in Fig. 7?

Line 333: You need to give some description, preferably in the instrumentation section, on how vertical velocities are measured.

Lines 345-349: Again, these are not considered high for midlatitude continental convection. See, for example, Wang et al. (2020): https://agupubs.onlinelibrary.wiley.com/doi/epdf/10.1029/2019JD031774.

Line 355: Earlier you mentioned that mass modes were used for discrimination. Mass size distributions (MSDs) do not peak at the same sizes as number size distributions (PSDs). Please clarify.

Line 356: Here again you say mass mode but you refer to a PSD. A mass mode would require some type of a mass-size relationship (used to integrate and derive LWC/IWC), unless independent bulk condensate measurements were provided.

Line 357: But did Costa et al. (2017) analyze a midlatitude convective system? These threshold values would likely be cloud-regime-dependent.

~Line 455: It is not clear to me how you make a distinction of "secondary ice", and you should mention other potential SIP mechanisms (e.g., drop freezing).

Line 457: This "typical PSD" is for Secondary Ice clouds? Fig. 13 sure looks like a small mode exists for Dp < 50 microns, in which case your argument is supported, but you state differently here.

Fig. 14: It's confusing to switch to Kelvin units here when you've been using Celsius for the rest of the manuscript. Recommend changing.

---

## Community Comment (CC1)

**Comment to Referee#1          19 June 2022**

We thank the Referee for helpful advice regarding manuscript acp2022-255: ***Process-based microphysical characterization of a strong mid-latitude convective system using aircraft in situ cloud measurements.***

The manuscript presents a case study of in situ cloud measurements within a European mid-latitude convective system or system cluster, which covered almost the entire Alpine region. The meteorological situation leading to such an extended cluster of deep convection is not frequent and the opportunity to perform measurements with cloud probes aboard a research aircraft is unique in this context. In situ measurements at altitudes between 2 and 12 km in an Alpine convective cluster require an aircraft, suited to fly in turbulent conditions and to avoid icing, that is why airborne in situ measurements with research aircraft like HALO are a challenge. This also points to the importance of this study. In order to sharpen the validity of this study the authors will pinpoint the geographic region it is related to, without generalizing it to mid-latitude convective Systems.

We thank the referee for suggestions to include further analyses and plots directly relating vertical velocities to particle size distributions and discuss the humidity. We see the importance of these correlations and will include them in the revised version of the manuscript. Nevertheless, we want to draw the referee's attention to several sections and figures in the manuscript, where some important and requested data is already shown. E.g. Fig. 10 shows time series of vertical velocities in each flight segment and the results are discussed in the text. In the analysis of microphysical quantities, LWC and IWC are related to other properties, such as MVD and Nc (Fig. 8). LWC and IWC are derived directly from particle size distributions using existing relations, and therefore are consistent within the other microphysical cloud data sets. Afchine et al., 2018, show that the use of cloud data from under wing stations is less prone to particle enhancements occurring around the fuselage of the aircraft. Hotwires or Nevzorov probes were not available in wing stations in the current HALO instrumentation.
Also, a time series of relative humidity over ice and water is presented in Fig. 9 and is the subject of discussion in Sections 2.4 and 2.5. We nevertheless will elaborate more on these topics and include relevant plots or information, as suggested.
Contrary to the referee's comment, cloud particle images from the Optical Array Probe CIP are included in Figs. 11, 12 and 13 and, together with the presented particle size distributions, support conclusions to be drawn about the identified cloud processes.

We take the referee's hint in pointing towards in this study measured maximum vertical velocities of 4 ms$^{-1}$, in the light of a range of literature stating maximum velocities of 20 ms$^{-1}$, to further emphasize in the manuscript that the cited vertical velocity was measured aboard an aircraft. These are not to be confused with measured maximum vertical velocities encountered at the center of updraft cores using e.g. remote sensing methods. Without the awareness towards the different inherent measurement methods, capabilities, regimes and peculiarities, there is the risk to misinterpret the different measurements. The vertical velocities measured on the aircraft are amongst the highest measured on different research aircraft in the comprehensive in-situ cloud climatology from Krämer et al., (2017).

We thank the referee for drawing our attention towards Giangrande et al. (2013) (https://journals.ametsoc.org/view/journals/apme/52/10/jamc-d-12-0185.1.xml), which will be referenced in our next draft. According to Fig. 1 (Giangrande et al., 2013) velocities of 20 ms$^{-1}$ are only reached within the updraft core, whereas over large parts vertical velocities remain below 4 ms$^{-1}$.

Also, Fig. 3 in Giangrande supports this conclusion. Again, one has to be aware that the underlying data set is based on aircraft in situ measurements and not remote sensing or drop sonde measurements. We will also clarify the meteorological situation and the position where the aircraft was flying with respect to convective cores in the manuscript.

[Figure]

*Figure 1 from Giangrande et al.(2013).*

[Figure]

*Figure 3 from Giangrande et al.(2013).*

Flight regulations under which HALO, a Gulfstream 550, operates are the same as commercial aircraft (EASA CS 25; there is no experimental class in German air law) and such is not a dedicated "storm-hunter" compared to a rugged P3 Orion or Hercules (used in the US). Thus, flights in known icing conditions, as well as flights outside the general g-load envelope are prohibited, just to mention two relevant limitations.

All in all, in situ measurements of cloud microphysical properties have been performed at high updraft speeds (but not at the highest vertical velocities, for reasons of flight control and safety), in a strong Alpine convective cluster system that in its magnitude and size is infrequent to this region in Europe. The presented analysis is a case study, based on a unique data set. Our study identifies and analyses cloud processes under these special meteorological conditions in an extended convective cluster. It further can serve as a unique reference data set calling for process model studies and intercomparisons to different measurement methods. One future goal is the evaluation of remote sensing methods e.g. polarimetric radars with respect to dedicated cloud properties. As such we believe that this study adds value and is of relevance for the scientific community.